# Herpesviruses in Head and Neck Cancers

**DOI:** 10.3390/v12020172

**Published:** 2020-02-03

**Authors:** Mikołaj Wołącewicz, Rafał Becht, Ewelina Grywalska, Paulina Niedźwiedzka-Rystwej

**Affiliations:** 1Student of Genetics and Experimental Biology, Student’s Circle of Immunology, Faculty of Mathematical, Physical and Natural Sciences, University of Szczecin, Felczaka 3c, 71-412 Szczecin, Poland; mikolaj.wolacewicz@gmail.com; 2Clinical Department of Oncology, Chemotherapy and Cancer Immunotherapy, Pomeranian Medical University of Szczecin, Unii Lubelskiej 1, 71-252 Szczecin, Poland; rbecht@pum.edu.pl; 3Department of Clinical Immunology and Immunotherapy, Medical University of Lublin, Chodźki 4a, 20-093 Lublin, Poland; ewelina.grywalska@gmail.com; 4Institute of Biology, University of Szczecin, Felczaka 3c, 71-412 Szczecin, Poland

**Keywords:** herpesviruses, oncogenic viruses, head and neck cancers

## Abstract

Head and neck cancers arise in the mucosa lining the oral cavity, oropharynx, hypopharynx, larynx, sinonasal tract, and nasopharynx. The etiology of head and neck cancers is complex and involves many factors, including the presence of oncogenic viruses. These types of cancers are among the most common cancers in the world. Thorough knowledge of the pathogenesis of viral infections is needed to fully understand their impact on cancer development.

## 1. Introduction

Head and neck cancers (HNCs) are diseases that can affect cells in many parts of the head and neck, including the paranasal sinuses, nasal cavity, mouth, throat, larynx, salivary glands, and thyroid [1]. Cancer is a disease caused by accidental errors in the replication process, by errors in the process of DNA repair, or by the body’s exposure to one or many physical and/or chemical oncogenic factors. All of these factors contribute to the occurrence of cytogenetic changes caused by a series of mutations of individual genes, resulting in uncontrolled cell proliferation [2,3,4]. In 2018, the Global Cancer Observatory (GCO) estimated there were 354,864 cases of lip and oral cavity cancers, 296,851 cases of brain and central nervous system cancers, and 41,799 Kaposi sarcoma cases, of which 177,384, 241,037, and 19,902 cases were fatal, respectively [5]. HNCs are mainly diagnosed in patients aged 45–50 [2,6]. However, since 1915, there has been an upward trend in the incidence of head and neck cancers among young people [2,6,7,8]. This is indicative of the high correlation of these types of cancers with environmental factors, smoking, alcohol consumption, diet-related factors, oral hygiene, and viral infections. In addition to factors such as alcohol and cigarettes, which are considered to be the main cause of about 75% of all cases of oral squamous cell carcinoma (OSCC), viruses are an important factor. The most common viruses related to HNCs that are involved in cell transformation are the human papilloma virus (HPV) [9,10], herpesviruses, adenoviruses [11], and hepatitis C viruses [12,13].

It has been shown that herpesviruses have an extreme ability to exploit a number of immune evasion strategies, leading to the possibility of them persisting in a latent state for a long period of time [14]. Some herpesviruses are known to be oncogenic and may have a potential role in cancer development. Of all the members of the extended herpesvirus family (Figure 1), only the roles of HHV-4 (the Epstein–Barr virus (EBV)) and HHV-8 are widely known, while the potential impacts of other herpesviruses (HHV-1/2, 3, 5, 6, and 7) are more disputed: The number of papers confirming the role of these viruses is limited, and based on their results, it can only be stated that these viruses are potentially oncogenic.

## 2. HHV-4/EBV (the Epstein–Barr Virus): Undoubtedly the Most Oncogenic Herpesvirus

The Epstein–Barr virus is etiologically linked to various malignancies, and over 90% of the adult population is infected with EBV, usually asymptomatically [16,17]. Interestingly, EBV-associated malignancies have a geographically specific pattern of occurrence [18]. It has been proven that the virus has the ability to infect both B lymphocytes and epithelial cells throughout the body, including in the head and neck region [19]. It was the first virus found to be directly related to carcinogenesis, and it has been classified as a group 1 carcinogen [20].

While EBV infection in oropharyngeal mucosal sites appears to be part of the virus–host interaction, the degree to which EBV infects the nasopharyngeal epithelium is surprising [21]. EBV is consistently detected in nasopharyngeal cancers [22]. Studies have shown that EBV DNA is clonal, which may be evidence of a relationship between nasopharyngeal cancer development and viral infection [23]. Serological tests in patients with nasopharyngeal cancer have shown an increased number of IgA and IgG antibodies against viral capsid antigens, early antigens, and nuclear antigens [24]. EBV gene expression in malignant NPC cells is restricted to *EBNA1*, *LMP1*, and *LMP2*. Thus, the augmentation of cellular immunity to EBNA1 and LMP antigens may be a potential alternative to chemo- and radiotherapy [25]. It has been shown that in some circumstances, an inflammatory environment can promote oncogenic change through the trafficking of lymphocytes into the nasopharynx, leading to an increase in Epstein–Barr virion reactivation in the B reservoir [21].

Samples from patients with oral squamous cell carcinoma (OSCC) in the oral cavity have been analyzed for the presence of HPV, HSV, CMV, and EBV viruses: EBV was identified in 57.5% of the examined persons. In addition, 7.5% of patients were identified as having the HHV-1 virus and 10% as having CMV. Coinfection with viruses was found in 30% of cases, with EBV and HPV coinfections occurring the most frequently [26]. In the same study, it was also mentioned that the involvement of EBV in OSCC probably depends on the geographical region, with the highest frequency noted thus far occurring in Southeast Asia [26]. Moreover, a positive correlation between different grades of OSCC and EBV DNA positivity has been confirmed, where the percentage of positivity for EBV increases from well-differentiated OSCC to poorly differentiated OSCC [27].

Studies have shown the presence of EBV in patients with Hodgkin’s Lymphoma (HL), with the highest prevalence in classic HL [25,28,29]. In HL, the viral genes EBNA1 and LMP-1 are expressed, and they play an important role in the transformation of infected B cells [25]. Moreover, T cells specific for the viral genes EBNA1 and LMP-1 lose their function during acute HL, and increased susceptibility to immunomodulatory molecules such as galectin-1 has been observed [25]. In addition, a relationship between the virus and the heterogeneous Non-Hodgkin’s Lymphoma (NHL) group (Burkitt’s lymphoma, NK-cell lymphoma, and proliferative lymphoma) has been demonstrated [30,31]. In Burkitt’s lymphoma, cells display an EBV latency I profile, expressing only EBNA1 [32]. It has been proven that the virus can cause proliferative diseases in people with immune disorders [33]. It is also believed that EBV plays an important role in the lymphomagenesis process [34]. It may also be associated with proliferative disorders in atopic lymphocytes and cancer cells after transplantation [35]. Further, its association with lymphomas in patients with immunosuppression and angiocentric lymphomas in the paranasal sinuses indicates a possible causal role of EBV in these forms of NHL [36].

A link has also been found between plasmablastic lymphoma (PBL) and the presence of EBV, where 9 out of 15 patients showed abundant EBV-encoded nuclear RNA transcripts in the absence of EBNA-2 [37]. In addition, five of the EBV-positive cases expressed LMP-1 [37]. No effective and standard PBL treatment has been found thus far: Treatment includes chemotherapy and occasional radiation therapy [19].

The presence of EBV has also been confirmed in the development of Burkitt’s lymphoma (BL) [19]. Control tests in patients with Burkitt’s lymphoma showed a higher level of EBV antibodies than in healthy people [19]. The data suggested that BL patients with EBV may also be coinfected with *Plasmodium falciparum*, as malaria is thought to diminish T-cell control of proliferating EBV-infected cells [19]. Current evidence has indicated that EBV is an important pathogenic agent during the development of Burkitt’s lymphoma (mainly in African countries, where malaria is a pivotal cofactor) [36].

PCR-analyzed and DNA-sequenced samples from Sudanese snuff (toombak) users and nonusers showed that EBV was present in 65% of samples from toombak users. In addition, the virus was present in 84% of nonusers [38]. The presence of EBV proteins and DNA has also been found in most patients with natural-killer-cell lymphomas. These lymphomas have a poor prognosis, even when timely diagnosed [19]. At present, many studies have clearly shown a correlation between the occurrence of head and neck cancers and the presence of EBV [19]. Studies have also shown that there is no specific relationship between smoking or alcohol consumption and the incidence of cancer [19,38]. However, there has been evidence demonstrating that coinfection with various viruses (mainly HPV and EBV) has an effect on cancer development [19,26,39]. Further research is needed to determine the potential role of EBV and the possible significance of HHV-1 as a coinfection factor in oropharyngeal cancer [26].

Some data have shown a genetic cause for higher risk of nasopharyngeal carcinoma: *RASSF1A* and *p16* inactivation, which occurs early on in the pathogenesis of this type of carcinoma, may predispose individuals to subsequent EBV infection that originates from lymphoid tissues and circulating B cells [40].

An important factor in understanding the molecular pattern of EBV and its impact on cancer development is the relationship between p53—a protein that regulates the cell cycle—and apoptotic cell death [41]. Interactions between p53 and EBV oncoproteins have been observed in many types of cancers, including head and neck cancers, and the concentration of p53 also determines cell cycle arrest and apoptosis in EBV-infected B cells [42]. It has been suggested that a specific isoform of p53 may be characteristic in patients with HNCs [41,42]. Moreover, an elevated level of p63 is associated with a prognosis of nasopharyngeal carcinoma, which may draw attention to a completely new method of diagnosing this cancer [43]. It also has been proven that EBV factors may manipulate the host epigenetic machinery or act in a “hit and run” manner, meaning that EBV participates in the early stages of tumor development by initiating oncogenic changes within the cell, but then disappears [44].

## 3. HHV-8/KSHV (Kaposi Sarcoma Herpes Virus): The Second Most Widely Distributed Oncogenic Herpesvirus

HHV-8, which was originally discovered in Kaposi sarcoma (KS), is associated with around 1% of all human malignancies and is classified, together with HHV-4, as a class I carcinogen [32]. In some areas of Africa, more than 70% of the population is HHV-8-seropositive, making the virus an important oncogenic factor. KS is usually associated with HIV/AIDS, but it cannot transform cells in culture and does not sustain itself without EBV coinfection [45]. In some lymphomas, EBV and HHV-8 coinfect tumor cells in 90% of cases [45]. Despite the many successes in the diagnosis and treatment of KS [46,47], the pathogenesis process itself remains unexplained [46].

HHV-8 has been linked to many diseases, including B-cell lymphoproliferative disorders and multicentric Castleman’s disease, which can progress into KSHV-associated non-Hodgkin’s lymphoma and also, primary effusion lymphoma (PEL) [32]. In vivo studies in mice have confirmed that HHV-8/HHV-4 dual-infection enhances HHV-8 persistence and tumorigenesis, and some authors have claimed that this may be a rule in lymphoproliferative disorders [32].

In larynx cancer, the presence of HHV-8 DNA has been detected using PCR techniques. The presence of the virus was confirmed in two samples from both sick and healthy people. Due to these results, no significant relation was found between the occurrence of larynx cancer and KSHV infection [48].

There have been studies on the presence of infectious HHV-8 in the saliva of patients with Kaposi sarcoma, which could be indirectly related to head and neck cancers and the HHV-8 virus, but the potential and importance of salivary shedding in HHV-8 transmission have still not been determined [49].

## 4. Other Potentially Oncogenic Herpesviruses with an Impact on Head and Neck Cancers

The potential role of HHV-1 in head and neck cancers has been described in some papers. Nevertheless, the impact of this virus (or lack thereof) is not unequivocal and has not been confirmed by in vivo observations.

It has been shown that patients with head and neck squamous cell carcinoma (HNSCC) are often coinfected with HHV-1, but the infection is usually asymptomatic [6,50]. Higher HHV-1 shedding in patients treated for HNSCC is considered to be due to the high level of stress associated with hospital procedures and healing trauma [51]. Due to the several molecular mechanisms that have been observed during HHV-1 infection that affect apoptotic pathways by downregulating p53, causing interactions with DNA repair mechanisms, and chromosomal instability, it has been suspected that HHV-1 may affect the radiation response of infected cells during HNSCC treatment [6]. In vitro studies on cell line UD-SCC-2 have shown [6] that HHV-1 infection modulates the radioresistance of HPV16-positive hypopharyngeal carcinoma cells. As it is known that HHV-1 may coinfect HPV-infected premalignant or malignant cells, it is crucial to know if HHV-1 infection can impact HPV-infected cell survival [51]. Research by Turunen et al. [6] has confirmed that the main roles of HHV-1 in HNSCC are inhibiting the intrinsic apoptotic pathway using the proteins ICP-0, Us3, and Us5 and lowering HPV-specific antiapoptotic gene expression in infected cells.

In another study concerning oral mucositis (OM), which is a side effect of antineoplastic treatment in patients with HNSCC, it was shown [52] that the presence of the HHV-1 and HHV-2 viruses was not correlated with the existence of OM: Despite this fact, the seroprevalence of IgG was 97.8%.

There is also evidence that HHV-1 is associated with oral squamous cell carcinoma (OSCC) [53,54]. A study in Poland, which was performed on freshly frozen tumor tissue fragments from 80 patients with OSCC, showed that in 7.5% of the samples, HHV-1 was present: Thus, HHV-1 may be, according to the authors, a potential coinfector in OSCC and an important cofactor in oropharyngeal cancer [26].

HHV-1 and HHV-2 are also understood by some researchers to be a potential risk factor for head and neck carcinomas, together with tobacco, alcohol, and oncogenic HPV [55]. This was also confirmed in a Swedish study performed on patients with oral cancer [56] and in an American study in which it was suggested that HHV-1 may enhance the development of OSCC in individuals who are already at increased risk of the disease due to cigarette smoking or HPV infection [57].

On the other hand, there have also been papers that have not confirmed the correlation between herpes virus infection and the risk of head and neck cancers, but those studies were more focused on HHV-2 infection. Maden et al. [58] found a nonsignificant level of risk for oral cancer associated with HHV-2 infection. In addition, there was no significant association between HSV1/HSV2 infection and head and neck cancers in a study by Parker et al. [59]. Similarly, the results were insignificant in patients with OSCC and in toombak and non-toombak users [38]. Moreover, no correlation between OSCC development and the presence of HHV-1 has been detected [60]. However, the question of whether HHV-1 and HHV-2 play an active role in head and neck cancer development or whether they are just local bystanders in the immune-deficient area around a tumor remains unanswered.

It has been shown that the oncogenic mechanisms exploited by these viruses may mainly involve antiapoptotic activity: In HHV-2, this includes the ICP10PK protein, which inhibits apoptosis through the activation of the Ras/Raf-1/MEK/ERK pathway [61].

In summary, it can be stated that the role and impact of HHV-1 and HHV-2 on head and neck cancers remain unclear. Efforts are being made to explain the crosstalk between these factors, with a special emphasis on possible treatments. One promising therapy may be the use of the oncolytic herpes simplex virus NV1020, which is an effective model in mice that causes tumor regression [62].

Reports on HHV-3 impact on head and neck cancers have been very limited but have shown that the virus may not be an oncogenic factor in the etiology of head and neck cancers. One of these papers is a description of a case concerning a patient with OSCC with negative PCR results for VZV but with a serological test that was VZV-positive. This 64-year-old man had a history of stage IV oropharyngeal squamous cell carcinoma treated with cisplatin and cetuximab, followed by radiotherapy. Unfortunately, the patient died 5.5 weeks after starting VZV treatment [63]. This was a case of a temporal relationship between VZV and myelitis in an immunocompetent host, which is rather rare. Nevertheless, on the basis of this report, it was stated that VZV infection should be included in the diagnosis of OSCC. Data have shown that HHV-3 acts by activating AP1, which is a transcription factor upregulating cellular proliferation [64]. The puzzle behind the long persistence of the HHV-3 virus in a host may also be connected to apoptosis inhibition caused by the IE63 protein [65], which is a major viral latency protein that is required for the inhibition of an alpha-interferon-induced antiviral response [66]. There was one cohort study on a large patient group with herpes zoster that was performed to identify whether a diagnosis of VZV is a risk factor for a subsequent malignancy [67]. The results showed that the incidence of cancer was significantly greater among patients with VZV than among those without herpes zoster, and lymphoma was the most frequent cancer that occurred [67]. The conclusion was that clinicians should be alert for malignancies in patients with VZV.

Although CMV is considered to be a non-oncogenic virus, many clinical and experimental findings have suggested the partial contribution of CMV to malignancy and chemoresistance in tumor cells infected by different entities [68]. In a paper from many years ago [69] on patients with nasopharyngeal carcinoma in North Africa, the presence of CMV was confirmed. An analogous study was recently performed in Sudan, and CMV was identified in almost one-third of the samples, which indicated a relatively considerable association between CMV and nasopharyngeal carcinoma [70]. In addition, some observations were made in patients with head and neck cancers who were receiving radiotherapy or radiochemotherapy to determine the impact of CMV infection on the risk of death [71]. It was concluded that the risk of death in patients who tested positive for CMV in their plasma was significantly higher than in patients with no CMV detected in their plasma, so diagnosing and treating these viral infections is crucial for better cancer treatment [71]. It has been shown that CMV causes an oncogenic effect through several mechanisms, including inducing cell cycle progression, chromosomal aberrations, and VEGF expression, activating cell motility and migration, and inhibiting DNA damage repair and apoptotic pathways [72].

There is evidence that CMV can be transmitted to tumor cells and may play an important role in modulating the tumor microenvironment by either inhibiting or promoting the tumor cells themselves [73]. Therefore, CMV may be a potential therapeutic target in HNCs.

As for HHV-6 and HHV-7, it has been previously confirmed [74] that these viruses exist in salivary glands, which are a reservoir for HHV-6 and HHV-7, but whether they have an impact on HNCs is questionable. In one paper on salivary gland neoplasm, it was confirmed that apart from the presence of these viruses in salivary glands, no correlation existed between HHV-6 and HHV-7 and salivary gland neoplasms [75]. In another study that detected HHV-6 genotypes through the in situ hybridization of carcinoma biopsies from the oral cavity, salivary glands, larynx, breasts, and cervix, the hybridization signal was strong enough to conclude that HHV-6 possesses the tumorigenic potential and demonstrates virus-transactivating properties [76]. The virus was found in the cytoplasm, cell membrane, and nucleus, but not in the cells surrounding the carcinomas [77]. In contrast, relatively low levels of HHV-6 were detected in a study by Yadav et al. [78]. Nevertheless, the conclusion was drawn that this virus, along with other carcinogens, may be an important factor in oral carcinomas. Moreover, the lack of an unequivocal answer about the existence and role of HHV-6 in oral squamous cell carcinoma may be indirect proof of the fact that this virus is involved in a low percentage of cases or that it uses a “hit and run” mechanism [79]. Studies on the role of HHV-6 in other carcinomas have shown that this virus may act by causing changes in apoptosis, in cellular transformation, in invasion and proliferation pathways, and in the transactivation of other viruses (mainly EBV, HPV, and HHV-8), which is convincing enough to merit taking a closer look at the role of these viruses in HNCs [80]. Other studies have also shown that the direct oncogenic mechanism exploited by HHV-6 is cell cycle arrest in the G2/M phase [81]. It is also crucial to identify chromosomally integrated HHV-6 (ciHHV-6), as high DNA viral loads of this virus in patients may be misleading in terms of diagnosis due to the fact that integration into the human chromosome is an oncogenic viral mechanism [82].

## 5. Conclusions

Of the members of the *Herpesviridae* family, HHV-4 and HHV-8 are the viruses with a proven oncogenic impact on many malignancies, including head and neck cancers. Other herpesviruses may also potentially play a role, and the involvement of these viruses in head and neck cancers was briefly discussed. Clearly, further research is needed (especially research involving molecular and in vivo trials) to be able to draw detailed conclusions on the impact and role of herpesviruses in head and neck cancers.

## Figures and Tables

**Figure 1 viruses-12-00172-f001:**
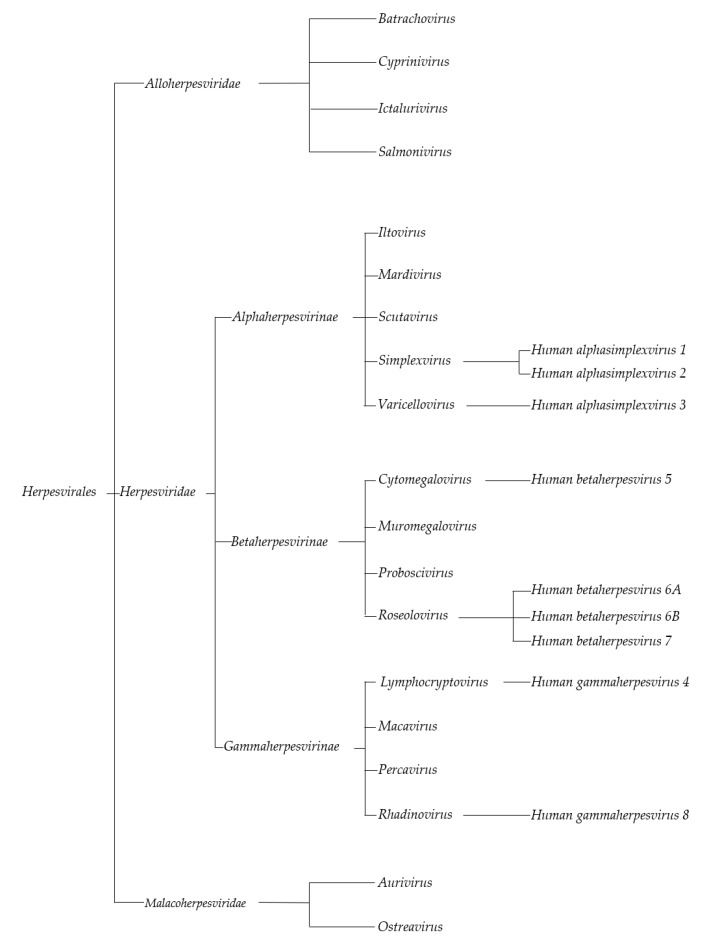
Systematic tree of Herpesvirales order (based on The International Committee on Taxonomy of Viruses) [15].

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
