# Peer review of "Herpesviruses in Head and Neck Cancers"

_viruses, 2020, doi:10.3390/v12020172_

Round 1
Reviewer 1 Report
The authors propose an overview of the relationship between herpesvirus infection and risk of oral cancer. They discuss discrepancies found in the literature. The manuscript is suitable for publication in my opinion. I advise English proofreading by a native speaker if possible.
Author Response
Thank you for the review. We would like to inform, that the manuscript undergone English editing.
Regards,
Reviewer 2 Report
The authors have not adequately addressed my previous concerns. The only human herpesviruses clearly associated with cancer are EBV and HHV-8. The remining viruses (HSV-1 HSV-2, VZV, CMV, HHV-6, HHV-7) are not considered oncogenic and the authors continue to include very weak papers that often rely on PCR (not identifying the cell type infected) and serology.
Author Response
After consulting academic editor, we restructured the manuscript not to leave any doubts, that only HHV-4 and HHV-8 are clearly associated with cancer. We introduced different chapters to emphasize this fact. All the remaining viruses were clearly marked as only potentially oncogenic and some of the studies are based only on PCR reactions or serology. All the changes in the text are marked in red colour. Therefore, we would like to kindly draw the Reviewers attention to the fact, that showing this kind of results was intended only to present the available data from the literature. Also, the paper has undergone English editing, that hopefully makes it better in overall reception.
We would like to kindly as to consider the corrected form of the manuscript again.
Kind regards,
This manuscript is a resubmission of an earlier submission. The following is a list of the peer review reports and author responses from that submission.
Round 1
Reviewer 1 Report
Dear authors,
The manuscript submitted by Wołącewicz et al.entitled “Herpesviruses in head and neck cancers”, although addressing an important objective has several and serious flaws and mistakes. Accordingly I recommend that it should be rejected.
Reviewer 2 Report
This manuscript is a review of the role of human herpesviruses in head and neck cancers. The review covers all eight human herpesviruses with a focus on HSV1,2 and EBV.
The main criticism of this review is that the authors provide little detail on each virus. They give one or two sentence summaries of published papers, but did not provide any real insight or description of the referenced work. As a result there is little real information given in the review. For example, in the section of HHV-1 the authors reference several times, reference 6, stating that HHV-1 infection modulates radio-resistance of cancer cells. Their statement infers that HHV-1 infection of cancer cells from patients are more resistant to radiation therapy compared to cancer cells not infected with HHV-1. What they fail to point out is that this is an in vitro study only. It does not deal with actual cancer cells isolated from a patient that are also HHV-1 infected. They don't provide any discussion about whether or not studies have demonstrated actual cancer cells that are co-infected with HHV-1. As a result, this part of the review is misleading as the referenced work doesn't address real-life situations with HNCC that are known to be co-infected with HSV. This type of error is present throughout the study and significantly diminishes the enthusiasm for the review..